# Predictive Value of the Münchener Funktionelle Entwicklungsdiagnostik Used to Determine Risk Factors for Motor Development in German Preterm Infants

**DOI:** 10.3390/biomedicines11102626

**Published:** 2023-09-25

**Authors:** Anna Janning, Hanne Lademann, Dirk Olbertz

**Affiliations:** 1Department for Pediatrics, Universitätsklinikum Würzburg, 97070 Würzburg, Bavaria, Germany; 2Department of Pediatrics, University Rostock, 18057 Rostock, Mecklenburg-Vorpommern, Germany; hanne.lademann@gmail.com; 3Department for Neonatology, Klinikum Südstadt Rostock, 18059 Rostock, Mecklenburg-Vorpommern, Germany

**Keywords:** bayley scales of infant development, developmental care, Münchener Funktionelle Entwicklungsdiagnostik, neonatology

## Abstract

Early diagnosis of developmental delays is essential to providing early developmental care. The Münchener Funktionelle Entwicklungsdiagnostik (MFED) is a simple and cost-effective tool for diagnosing the development of infants and young children. Nevertheless, the MFED has not been a well-studied part of current research. This retrospective cohort study aims to detect risk factors and assess the impact of developmental care during the first twelve months of life, using the MFED. Furthermore, it determines the MFED’s predictive value by comparing results with an international gold standard, the Bayley Scales of Infant Development II (BSID II). The study included 303 infants born between 2008–2013 in Rostock, Germany, with a birth weight of ≤1500 g and/or a gestational age of ≤32 weeks, who were evaluated with the MFED at twelve months of age. To ascertain the predictive value, 213 infants underwent BSID II assessment at 24 months of age. Intraventricular hemorrhage (IVH), necrotizing enterocolitis (NEC), and periventricular leukomalacia (PVL) were significantly associated with a higher risk of developmental delay across various domains. Post-discharge developmental care therapies did not indicate any clear beneficial effect on the infant’s development. Nevertheless, some domains of MFED demonstrate predictive value, warranting increased attention for this diagnostic.

## 1. Introduction

Modern medicine provides a high level of safety for both mothers and newborn infants. However, premature infants require special care to enable them to live an unimpaired life. The preterm birth rate in Germany and Europe is low compared with other continents, standing at approximately 8.5% [1,2]. Premature birth has various causes. Multiple birth has been identified as a significant risk factor for premature birth in Germany, as revealed by a large study published as part of the Child and Adolescent Health Survey (KiGGS), taking non-medical factors into consideration [3]. In addition, medical conditions such as infections, premature rupture of membranes, and maternal diseases like hypertension or diabetes should be regarded as risk factors [4,5]. Steadily improving care for preterm infants is leading to ever-decreasing rates of mortality and disability among survivors [6]. However, especially very preterm infants are vulnerable to preterm-associated diseases. Due to the immaturity of various organ systems, premature infants may develop various diseases, such as bronchopulmonary dysplasia (BPD), necrotizing enterocolitis (NEC), or retinopathy of prematurity (ROP). In conjunction with low birth weight, sepsis, and non-prematurity-related factors, a significant risk of motor, cognitive, and somatic developmental delay is associated with these disorders [7,8,9]. This is evidence that earlier birth may increase the risk of impairments [10]. Additionally, it is important to consider the children’s diet and its influence. Our research group has already published results on the cognitive, motor, and somatic development of 24-month-old preterm infants. These results show that exclusive formula feeding is a clear risk factor for cognitive and motor development delay, and BPD has a poor effect on length growth [11]. Although other studies have also demonstrated a positive effect on catch-up growth with formula feeding, effects on neurological development have not yet been described [12,13].

There are various approaches to preventing these risks. In addition to attempting to delay preterm birth, the World Health Organization (WHO) has published a list of recommendations to enhance the development of preterm infants. This includes kangaroo care as well as administering surfactant or oxygen for appropriate indications [14]. Another important aspect is the provision of follow-up and developmental care. The follow-up of preterm infants is regulated in various ways internationally. Germany has a national guideline prescribing a developmental test at two years of corrected age [15]. Moreover, there are recommendations on diagnoses and therapies up to the age of five [16]. The National Institute for Health and Care Excellence (NICE) in the United Kingdom has provided a guideline since 2017, outlining risk factors and criteria for developmental support, including gestational age <30 weeks or <36 weeks with a concomitant risk factor, as well as treatment recommendations [7]. In contrast, there are no standardized guidelines in the United States (US), with individual programs from various agencies [17]. One such program is the Newborn Individualized Developmental Care and Assessment Program (NIDCAP), which involves developing an individual care plan shortly after birth by analyzing behavior. No benefit has been proven so far [18].

To ensure optimal therapy for premature infants and avoid substantial costs for the health care system [17,19], timely diagnosis of developmental delays is essential. The Bayley Scales of Infant Development (BSID) are the gold standard for diagnostic purposes and are recommended by German guidelines. This test was developed by Nancy Bayley in 1969, and a revised second version was published in 1993. It includes a cognitive and motor scale that can give an indication of global developmental delay. The tasks, as well as the entry and exit levels, are described in concrete terms. The evaluation is completed by using a raw score, which can then be converted into the respective index. While less prevalent, the Münchener Funktionelle Entwicklungsdiagnostik (MFED) is a cost-effective alternative that requires less effort [20]. This test is based on data from a German reference cohort. It enables the isolated observation of individual developmental areas and is oriented toward the minimum norm of the reference cohort. The result of this test is a developmental age, which, when compared with the corrected age, can give an indication of a developmental delay. Studies on validation and predictive value have not yet been conducted. In general, there are indications that the results of developmental tests can also have positive predictive power for the future. For example, the General Movement Assessment showed good predictive power compared with the BSID, while the BSID III was unable to predict cognitive development from 24 months to four years [21,22]. 

The study aims to identify risk factors and the influence of therapies on motor development through the MFED. Additionally, the study seeks to determine the predictive power of the BSID II based on the results.

## 2. Patients and Methods

### 2.1. Study Design

This retrospective analysis is based on the data of all children born at the university women´s clinic, Klinikum Südstadt, Rostock, Germany, between 2008 and 2013 and treated in the neonatal intensive care unit (NICU). About 30% of the 3000 children born in the clinic each year require hospitalization. All children born with a birth weight of ≤1500 g and/or a gestational age of ≤32 weeks were included, which amounts to approximately 70 children per year. The NICU is a level III perinatal center with ten intensive care beds and dialysis capabilities. However, liver replacement, extracorporeal membrane oxygenation, and cardiac surgery are not available. Infants who died before discharge have been excluded. 

All newborns in the study had their somatic parameters, vital signs, and blood gas analyzed. Additional risk factors, diseases, and nutrition were recorded until discharge, which typically occurs at 36 weeks of gestation. Please refer to Table 1 for the detailed definitions of the considered risk factors. All children in this cohort received early developmental care in accordance with the hospital’s standard operating procedure (SOP) during their stay, which aligns with the national guidelines. This care included daily individual physiotherapy, support with sleep patterns, and the involvement of parents in the care process. The 20 min physiotherapy sessions were based on neurodevelopmental principles and aimed to prevent pneumonia, promote sensorimotor development, and provide orofacial stimulation. Parents were also involved in facilitating bonding with their children. The developmental care assessed in this study involved post-discharge physiotherapy and early education, which ideally should be provided weekly, although such provision is not standardized. The objective of early education is to promote the child's individual development and social integration. All infants included were also fed according to the hospital's internal SOP, as there were no national guidelines for feeding preterm infants at the time of the study. The infants were fed breastmilk or donated breastmilk whenever feasible. The categorization of “formula milk” or “human milk” was based on data obtained until the infants were 6 months old and performed by two unbiased peers.

### 2.2. Assessment

Based on the Federal Social Welfare Act, all premature infants are invited to attend consultations at three, six, nine, twelve, and 24 months, during which their somatic, motor, and cognitive development is examined. According to this legal regulation, all premature infants weighing less than 1000 g at birth are entitled to regular follow-up examinations, which are fully covered by health insurance. In line with the national guideline [16], our hospital extends these follow-up examinations to all premature infants. Legal requirements for follow-up examinations of preterm infants necessitate only a developmental test at 24 months utilizing the BSID. Additionally, it is recommended that a developmental assessment be carried out at twelve months employing a test of choice; in this study, the MFED was used as per the SOP. The MFED is structured around various functional areas and domains of cognitive and motor development. The motor development areas include crawling, sitting, walking, and grasping. It is assumed that children complete individual steps in each of these areas sequentially. Based on the Munich Pediatric Longitudinal Study (Münchener Pädiatrischen Längschnittstudie, 1971), tasks (items) were developed for each area that represents the minimum norm. In the reference cohort, 90% of the children were able to successfully complete these items. Subsequently, the obtained data can be used to calculate a developmental age for each area, which provides information about a possible developmental delay. The test commences with completing tasks of an age level lower than the child’s corrected age and finishes with the last level that the child was able to solve safely. The evaluation is performed by comparing the corrected and calculated ages. A difference exceeding two months is considered pathological and enables statements about either a partial or complete developmental delay. Up to a difference of two months, developmental control is necessary.

To determine the predictive value of the MFED, it is necessary to compare it with an established test procedure. In our case, the children were assessed with the BSID II at the age of 24 months. The BSID is a standard diagnostic procedure for assessing developmental progress in children up to the age of 42 months. It is divided into a Psychomotor Developmental Index (PDI) for motor skills and a Mental Developmental Index for cognitive development (MDI). The PDI evaluates gross and fine motor skills as well as postural control. In contrast to the MFED, an index (with a mean of 100 and a standard deviation of 15) is calculated to determine normative development. Normal development is considered to be within one standard deviation above or below the mean. A PDI below 70 indicates a severe developmental delay. The second edition was normed with US children in 1988 and translated into German [23] but was not normed with a German cohort. As the study focuses on motor development, only the PDI results were considered in the analysis.

### 2.3. Statistics

A descriptive analysis of the metric-scaled data, including weight, age at maturity, and somatic parameters, along with the prevalence of risk factors and gender, was conducted. The selection of considered risk factors was based on a literature review, resulting in 18 factors being considered. The risk factor definitions are presented in Table 1. All risk factors were recorded as “yes”, irrespective of their severity. The children were divided based on their completed tests. In order to analyze the risk factors at 12 months of age, data from all preterm infants examined with the MFED was utilized. For the calculation of the predictive value, we analyzed data from children with available results of both MFED and BSID II. 

Binary logistic regression was used to determine the influence of risk factors and therapies on the predictive value, resulting in an odds ratio (OR). An OR > 1 indicates an increased risk due to the respective factor, while an OR = 1 signifies no influence. The OR was used to determine the predictive value of delay in the BSID II given a delay in the MFED, with an OR >1 indicating increased risk for later developmental outcomes.

Significant risk factors were assessed using the Mann-Whitney U-test, and the impact of these risk factors on the achieved results was evaluated. In cases where the expected cell frequency was less than *n* = 5, a Fisher’s exact test was also applied.

Statistical analysis was conducted using IBM SPSS Statistics, Versions 25 and 27© [24,25], while graphical analysis was carried out with GraphPad Prism 5 and 9 [26,27]. 

### 2.4. Ethics

This study was approved by the ethics committee of the medical faculty at Rostock University (number of approvals: A 2015-0128, A 2015-0178, and A 2020-0207).

## 3. Results

### 3.1. Baseline Characteristics

418 out of 14,600 births met the inclusion criteria, of which 27 infants (6%) died before discharge. Table 2 displays the baseline data of the cohort. The genders were approximately distributed evenly, with no variations in somatic birth parameters or gestational age. 

The children in the cohort considered were born at an average of 29 ± 2 weeks of gestation, with three children (1%) being born below the limit of viability (<24 weeks). The average birth weight was 1202 ± 356 g. One-quarter of the sample was exclusively fed formula. Neonatal sepsis occurred in 15% (*n* = 45/303), while 17% (*n* = 51/303) developed BPD. Additionally, 15% developed IVH, with 4% being severe cases. In 33% (*n* = 100/303) of cases, a ROP diagnosis was made, whereas only 3% (*n* = 9/303) developed NEC (Table 2).

### 3.2. Assessment

303 out of 391 infants (72%) were examined with the MFED assessing motor development at twelve months of age. Of these, 213/303 (70%, Figure 1) also took part in the BSID II assessment at (corrected) 24 months. These datasets were utilized to ascertain the predictive value (Figure 1). 

Out of 303 infants, 86 (28%) infants did not exhibit delays in any developmental area. Therefore, on average, the overall calculated developmental ages did not differ from the corrected ages on average (crawl −0.24 months, sit −0.18 months, walk +0.07 months, grasp −0.16 months, Table 2). Between 6 and 13% (*n* = 15–34/303) of the infants were severely delayed, with a difference of ≥2 months. The impact of birth weight proved significant across all developmental areas. However, with an OR ≈ 1 and a small standard error, this finding lacked medical significance and was thus not considered in the subsequent analyses. 

The data suggest that several diseases linked to preterm birth heighten the risk of motor delay in various developmental domains. Especially among children with IVH, there appeared to be a significant increase in risk, estimated to be between 5 and 6 times higher (crawling OR 4.9; grasping OR 6.1; *p* < 0.05; Figure 2a,d). Furthermore, children with this condition demonstrated an average delay of approximately 1.3 to 1.8 months in motor development compared with those without. The motor delay was observed in 53–79% (*n* = 10–15/19) of the affected children, compared with 29–38% (*n* = 64–89/282) of the unaffected infants (*p* < 0.01, Figure 3a). 70–90% of preterm infants diagnosed with PVL demonstrated significant development delays across all four developmental domains in comparison to the unaffected cohort (Table 3). The two domains related to sitting and walking were impacted to a greater degree (OR 6.3–9, *p* < 0.12, Figure 2b,c and Figure 3c). Similar outcomes emerged for children with NEC and pneumonia. The crawling age was 13.1-fold more likely to be delayed following NEC (*p* < 0.05) while grasping delays were 6-fold more likely (*p* < 0.12, Figure 2a,d). Furthermore, the affected cohort displayed a significantly higher delay compared with the non-affected cohort (75%, *n* = 6/8 vs. 29–34%, *n* = 70–83/242, *p* < 0.05, Figure 3b, Table 3). Pneumonia was found to be associated with a 5.7-fold increase in the risk of motor delay in crawling (*p* < 0.05, Figure 2a). Furthermore, there was a twofold increase in the proportion of delayed children within the cohort with pneumonia (70%, *n* = 9/13) compared with those without (34%, *n* = 80/235, *p* < 0.05, Figure 3d, Table 3).

### 3.3. Developmental Care after Discharge

14/303 (5%) children did not receive developmental care. No significant difference was observed in any developmental domain between children who received developmental care and those who did not (*p* > 0.05, Table 4). All children in the cohorts with diseases that increase the risk of developmental issues received developmental care, except for those with pneumonia. However, regression analysis was not possible for this cohort due to the small sample size of *n* = 1/13.

### 3.4. Predictive Value

A prediction could be presented, particularly regarding the age of grasping. Preterm infants who experienced a developmental delay in this domain are at a 1.9 times higher risk of displaying a developmental delay in the PDI even at 24 months (*p* < 0.05). This was also a tendency for significance regarding the crawling age (OR 1.55, *p* = 0.05, Table 5).

## 4. Discussion

This study aims to identify risk factors and the impact of therapies on the motor development of very preterm infants by using the MFED. Additionally, the study validates the predictive value of the MFED concerning the BSID II. The study analyzed a cohort of 303 infants with very preterm birth and/or very low birth weight, which aligns with the baseline characteristics of German infants [28].

### 4.1. Baseline Characteristics

In contrast to global data, certain preterm-associated diseases were less frequent. Up to 39% experienced IVH worldwide, with 8–10% being severely affected (≥Grade III), compared with our cohort, which saw 15% (4% severely) [29,30,31,32]. ROP was observed in approximately 25% of cases, although most comparable studies only considered ROP ≥ grade II, leading to a lower prevalence in those reports [30,31,32,33]. A total of 17% of the infants included in our study were discharged with BPD. In international comparison, there is a wide range in the percentage of BPD cases (11–75%, [29,30,31,33,34]), which may be due to differences in the definition of BPD. Certain international studies vary in their inclusion criteria (gestational age < 27/<29 weeks [30,31,32]), which may result in increased rates of preterm-associated diseases. 

### 4.2. Assessment

Fortunately, on average, the infants in this study developed appropriately for their age. In contrast, van Iersel et al. evaluated 303 infants up to the age of twelve months by using the Alberta Infant Motor Scale (AIMS). This diagnostic tool measures the spontaneous movements of infants and is most commonly used at ages up to twelve months. The study demonstrated a developmental delay in preterm infants, whereas Pin et al. were unable to demonstrate a difference [32,35]. Consequently, the results indicate a risk of developmental delay in preterm infants.

Preterm infants with IVH are linked with a 5–7 times higher risk of experiencing a delay in motor behavior. This finding was also reported by Syrengelas et al. in their study of 403 preterm and 1038 term-born infants. The children were assessed using AIMS, and those with IVH had significantly poorer results than the term-born control group [29]. Moreover, lower birth weight does not seem to have an additional impact on this outcome [36]. Thus, IVH appears to have a significant impact on motor development.

Additional research is necessary for other identified risk factors, specifically pneumonia, NEC, and PVL. All of these factors showed a correlation with a high increase in the risk of a developmental delay in this cohort and although there was only a tentative significance, there may be an important medical impact. International studies have indicated that preterm infants with inflammatory diseases such as NEC or sepsis have significantly lower outcomes by the age of 18 months, as assessed by BSID III when compared with a control group [37]. This effect appears to be intensified with the presence of other diseases such as IVH, as demonstrated by Goldstein et al. in a study of 6638 preterm infants who suffered from both NEC and IVH. They published a risk increase by a factor of 6.3 for a motor delay when having the combination of those two factors [38].

Likewise, small studies investigated the impact of PVL on motor development. Despite the small number of participants, more than half of preterm infants suffering from PVL had significant motor delays [39,40].

Pneumonia was associated with an increase in risk by a factor of 5.7. There are no international studies published to compare with. Simply put, the administration of oxygen and ventilatory support appears to impact motor development [30,34]. Overall, extensive additional research is required.

### 4.3. Developmental Care after Discharge

In our study, there seems to be no positive effect of developmental care after discharge up to the age of 24 months. We already observed a similar result when we analyzed the somatic and cognitive development of this cohort [11]. It is challenging to compare our findings with the international literature, as there are no standardized guidelines or recommendations, leading to considerable heterogeneity. Soleimani et al. conducted a systematic literature review of 21 studies and demonstrated a positive effect on motor development at twelve and 24 months of age. However, the included studies focused solely on treatments during the inpatient stay, such as NIDCAP, and did not consider treatments after discharge [41]. Spittle et al. discovered a beneficial impact for treatments even after discharge, albeit only noticeable in motor development until toddler age, compared with cognitive development, which continued to have an effect until school age. Furthermore, there is significant variation in the follow-up programs among the examined studies [42]. In this study, we used the German guideline-based follow-up of very preterm infants [16], which was also applied to children with a higher gestational age and/or birth weight. The outcome of the children was equally positive compared with other German studies [43,44]. In the national context, a more precise analysis of developmental care and a larger cohort are necessary to classify and justify this positive result. Additionally, an analysis should be conducted to determine if the positive impact is evident in the subsequent development of the children. International studies suggest that developmental care is likely to have a positive impact.

### 4.4. Predictive Value

This study offers initial findings on the predictive value of MFED. Previous studies, such as those examining AIMS or BSID, found that early outcomes are indicative of later development for both cognitive and motor skills [33,45]. Thus, it is suggested that infants with developmental delays at twelve months of age receive intensive support. The MFED provides an advantage due to its affordability and ease of administration compared with similar tests. Additional research is necessary to assess the effectiveness of tailored support for premature infants with risk factors.

### 4.5. Limitations

The underlying test principles represent the greatest limitation of this study, as the MFED was standardized more than 40 years ago [18]. However, a new standardization is currently underway in a German study, which could provide a more accurate diagnosis. Additionally, this study utilized the second version of the BSID, which is no longer considered the gold standard. Studies comparing the two versions indicate that children assessed with the BSID II tend to receive a lower score and are more frequently diagnosed with deficits [46,47]. Additionally, it is important to note that the BSID II used in these studies is simply a translation and lacks standardization using a German reference group. The BSID III, on the other hand, is based on a German norm, and a comparison of these two norms revealed that German children perform significantly worse when assessed using the US norm than when assessed using the German norm [48]. In this context, the German standardization of the BSID is advantageous.

This study did not include maternal and socioeconomic factors. International evidence suggests that maternal education, as well as family socioeconomic status, may impact motor development [30,49]. The present lost-to-follow-up and retrospective study design should be taken into account when interpreting and comparing the data.

All in all, these limitations mean that additional research is necessary to validate and implement these findings. Nevertheless, this study establishes the groundwork for improved, harmonized follow-up care for preterm infants in Germany and may result in increased focus on the MFED as a simple and early diagnostic tool, a subject that will be explored in future projects.

## Figures and Tables

**Figure 1 biomedicines-11-02626-f001:**
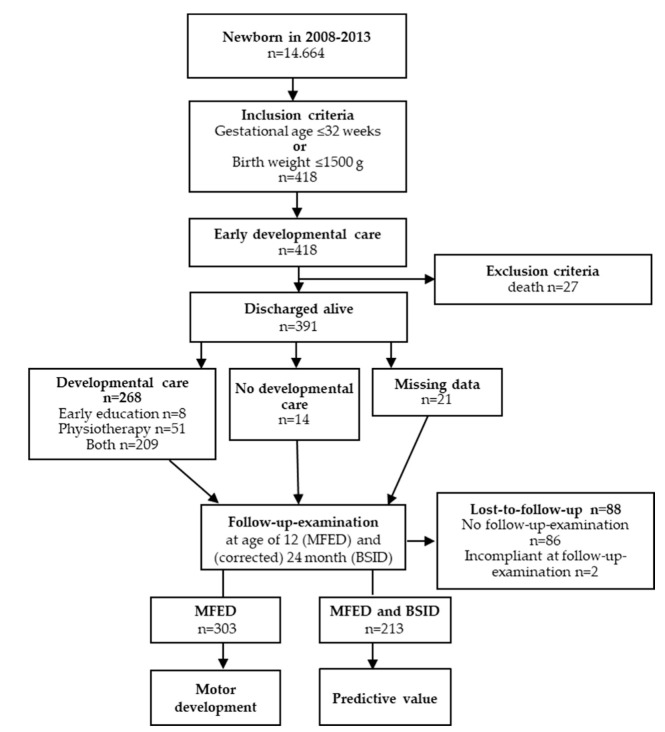
Study design. (Modified with permission from ref. [9]). A total of 418 infants (born between 2008 and 2013, with a birth weight ≤1500 g and/or gestational age of ≤32 weeks) met the inclusion criteria, while 27 infants were excluded due to death. Of the 418 infants, 268 received developmental care. An assessment of risk factors and the impact of therapies was conducted on 303 infants, as evaluated by the MFED. 213 of them were also evaluated by the BSID, and this information was utilized to examine the MFED’s predictive value. Abbreviations: *n* number, MFED Münchener Funktionelle Entwicklungsdiagnostik, BSID Bayley Scales of Infant Development.

**Figure 2 biomedicines-11-02626-f002:**
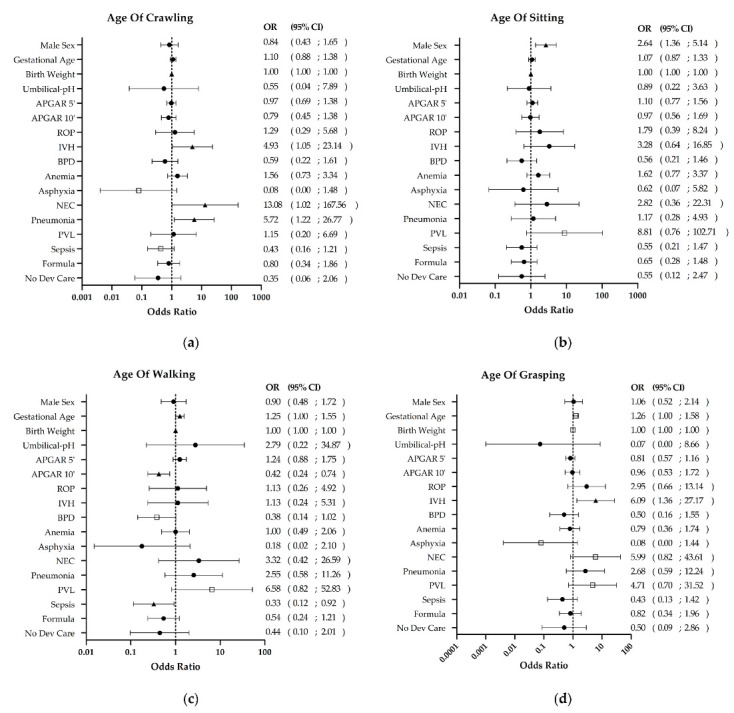
Risk factors for the MFED developmental areas in (**a**) Crawling, (**b**) Sitting, (**c**) Walking and (**d**) Grasping. IVH, PVL, NEC and pneumonia were found to have a significant impact on the risk of developmental delay, as determined by binary logistic regression analysis. Furthermore, it was found that children who did not receive developmental therapies did not have an increased risk of developmental delay. ● *p* > 0.12, □ *p* ≤ 0.12, ▲ *p* ≤ 0.05. Abbreviations: IVH intraventricular hemorrhage, ROP retinopathy of prematurity, BPD bronchopulmonary dysplasia, NEC necrotizing enterocolitis, PVL periventricular leukomalacia, OR odds Ratio, CI confidence interval.

**Figure 3 biomedicines-11-02626-f003:**
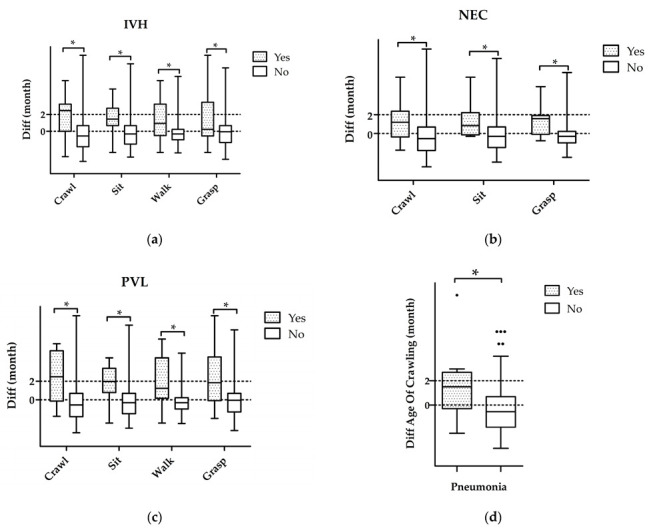
Developmental delays in preterm associated with (**a**) IVH, (**b**) NEC, (**c**) PVL and (**d**) pneumonia. Affected children have significantly higher developmental delays in at least one area. * *p* < 0.05. Abbreviations: Diff difference of the corrected and biological age, IVH intraventricular hemorrhage, NEC necrotizing enterocolitis, PVL periventricular leukomalacia.

**Table 1 biomedicines-11-02626-t001:** Definition of the risk factors under consideration.

Risk Factor	Definition
BPD	Need of oxygen 6 month after birth
NEC, IVH, PVL Anemia, asphyxia, sepsis, ROP	All grades
Formula feeding	never fed with breast milk
No Developmental care	Never receive physiotherapy/early education
Pneumonia	Pneumonic infiltrates in X-ray, clinical presentation, and elevated inflammatory markers

**Table 2 biomedicines-11-02626-t002:** Baseline characteristics.

		*n* = 303	
		*n*	%
**female**		154	51
**Gestational age** (weeks)	$\bar{x}\;$± sd	29 ± 2	
<24		3	1
24–27		89	29
28–31		183	60
≥32		28	9
**Birth weight** (g)	$\bar{x\;}$± sd	1202 ± 356	
<750		36	12
750–999		62	21
1000–1499		156	52
≥1500		49	16
**Formula feeding** until 6 months		75	25
**Sepsis** until discharge		45	15
**BPD** until discharge		51	17
**IVH** until discharge I.		24	8
II.		8	3
III. + IV.		13	4
**NEC** until discharge		9	3
**ROP** until discharge		100	33

Abbreviations: BPD Bronchopulmonary dysplasia, IVH intraventricular hemorrhage, NEC necrotizing enterocolitis, ROP retinopathy of prematurity, *n* number, sd standard deviation, $\bar{x}\;$
mean value.

**Table 3 biomedicines-11-02626-t003:** Developmental delay of the cohort assessed with MFED.

	Diff > 0 Mon		Diff ≥ 2 Mon		Average Diff
	*n*	%	*n*	%	$\bar{x}$ ± sd
**All** (***n* = 303/100%**)					
Crawl age	55	22	34	14	−0.2 ± 1.9
Sit age	88	35	16	6	−0.2 ± 1.6
Walk age	78	31	32	13	0.1 ± 1.8
Grasp age	61	24	15	6	−0.2 ± 1.4
**IVH ≥ grade II **(***n* = 19/6%**)					
Crawl age	3	16	10	53	1.8 ± 2.4
Sit age	10	53	5	26	1.4 ± 1.9
Walk age	2	11	8	42	1.5 ± 2.9
Grasp age	4	21	8	42	1.3 ± 2.2
**NEC** (***n* = 8/3%**)					
Crawl age	3	38	3	38	1.4 ± 2.3
Sit age	4	50	2	25	1.4 ± 2.1
Walk age	2	25	2	25	1.0 ± 2.7
Grasp age	4	50	2	25	1.4 ± 1.8
**PVL** (***n* = 10/3%**)					
Crawl age	1	10	6	60	2.3 ± 2.8
Sit age	4	40	5	50	1.9 ± 2.1
Walk age	3	30	5	50	2.4 ± 3.4
Grasp age	3	30	5	50	2.3 ± 2.9
**Pneumonia** (***n* = 13/4%**)					
Crawl age	3	23	6	46	1.5 ± 2.8
Sit age	5	39	2	15	0.9 ± 2.4
Walk age	4	31	3	23	1.0 ± 2.6
Grasp age	3	23	3	23	0.7 ± 1.8
**Formula-feeding** (***n* = 59/19%**)					
Crawl age	9	15	14	24	0.1 ± 2.4
Sit age	18	31	6	10	0.1 ± 1.9
Walk age	12	20	11	19	0.2 ± 2.2
Grasp age	13	22	6	10	0.1 ± 1.7

Abbreviations: diff difference, mon month, *n* number, IVH intraventricular hemorrhage, NEC necrotizing enterocolitis, PVL periventricular leukomalacia, MFED Münchener Funktionelle Entwicklungsdiagnostik, sd standard deviation, $\bar{x}$ mean value.

**Table 4 biomedicines-11-02626-t004:** Impact of developmental care therapies after discharge.

MFED		Physiotherapy/Early Education		OR (95% CI)	*p*-Value
		No (*n* = 14)	Yes (*n* = 268)		
Crawl age	$\bar{x\;}$ ± s	−0.2 ± 3.2	−0.3 ± 1.8	0.4 (0.1; 2.1)	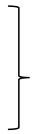 †
Sit age		−0.2 ± 2.9	−0.2 ± 1.5	0.6 (0.1; 2.5)	
Walk age		0.0 ± 3.0	0.1 ± 1.8	0.4 (0.1; 2.0)	
Grasp age		−0.3 ± 1.9	−0.2 ± 1.3	0.5 (0.1; 2.9)	

† not significant (*p* > 0.05). Abbreviations: OR Odds Ratio, CI Confidence interval.

**Table 5 biomedicines-11-02626-t005:** Predictive value of MFED.

	OR	*p*-Value
Crawl age	1.55	0.05
Sit age	1.31	0.35
Walk age	0.71	0.15
Grasp age	1.93	0.03

Abbreviation: OR odds ratio.

## Data Availability

Data available on request due to privacy restrictions.

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
