# Peer review of "Predictive Value of the Münchener Funktionelle Entwicklungsdiagnostik Used to Determine Risk Factors for Motor Development in German Preterm Infants"

_biomedicines, 2023, doi:10.3390/biomedicines11102626_

Round 1

Reviewer 1 Report

per attached file

This appears like a rough translation that requires considerable revising to make it clear to the reader.

Reviewer 2 Report

I have read this paper with great interest, and with a background on clinical research in neonatology, including studies related to outcome. I’m fully aware of the major efforts made to treat the all individual cases present in this decade cohort, and respect this.

After repeated reading of this paper, I have two types of concerns that I would like to share with the authors and the editorial office.

My first line of concerns relates to the topic and the current journal. Biomedicines has a specific focus, and this focus is almost absent in the current version of the paper. I therefore suggest the authors either to reconsider the targeted journal, or alternatively, provide more reflection and evidence to link with the aims of this journal (cf link on the homepage, aims and scope, to biomarker discovery and early diagnosis research. At present, the paper seems to have two ambitions, i.e. predictive value of MFED to Bayley results, and risk factor similarity.

My second line of concerns relate to the topic, and current description of the study, and these comments are topical, and are provided chronologically.

Abstract

MFED at ? 12 months, the next but last sentence is not supported by the current data or analysis, suggest to reconsider or remove, the last sentence, developmental areas: after reading the full paper, this seems to relate to the MFED, so recommend to rephrase this.

Introduction:

are you sure that multiplets are the most common cause of preterm birth in Germany ?

line 54: I assume prenatal steroids ? and the ‘textual link’ to follow up practices is not yet clear to this reviewer, so suggest to reconsider this and add a link

General: it seems that the MFED is almost exclusively focused on motor development (although the methods also suggest cognitive development), so do you link this with the ‘motor’ Bayley, or the psycho’Bayley ? this is not yet sufficiently clear. Furthermore, somewhat surprise that the version 2 is still used. This is a limitation and should be further stressed.

Methods

Not sure if the figure 1 is not more a result ?

We do need more clearer description on the what and how.

What do you mean with early developmental care (? Nidcap or similar ?), and developmental care (Bobath, or other ?, or not standardized), what ‘type’ of BSID has been reported (cfr higher)

Has the German (language) version fhte BSID version 2 never been normed ? (at least, we need a reference on the norming).

We need a definition for BPD, NEC (including Bell 1 ?) and ROP (likely any grade ?), how to define neonatal pneumonia is quite a challenge ?

Results

Formula feeding until 6 months ? what do you mean with this ? the absence of breastfeeding, or no introduction of other food until 6 months ?

I do not yet understand table 2: what do you mean with > or > Mon ?

Absolved is likely not an accurate wording, suggest to check

Table 3 relates to developmental care after discharge ? if so, suggest to add (similar for 4.3 subtitle likely)

Although there is a lot of debate on the relevance of p-values, but 0.05 is not significant.

Discussion

Has the cognitive development been considered in this analysis ? quite unclear to this reviewer.

Limitations, cfr higher: were postdischarge programs harmonized, and what has been provided ?

Round 2

Reviewer 1 Report

attached file

Substantially improved.  A few remaining corrections to be made.

Reviewer 2 Report

i suggest another revision, as this design does not allow any causality claim, but only associations. I therefore suggest the adapt the wording used (results in, causes, to is associated etc)

similarly, how 'robust' is the claim on the absence of any effect of the intervention (power, timing interventions pre- and postdischarge versus assessment time windows)
